# Higher Responsiveness of Pattern Generation Circuitry to Sensory Stimulation in Healthy Humans Is Associated with a Larger Hoffmann Reflex

**DOI:** 10.3390/biology11050707

**Published:** 2022-05-05

**Authors:** Irina A. Solopova, Victor A. Selionov, Egor O. Blinov, Irina Y. Dolinskaya, Dmitry S. Zhvansky, Francesco Lacquaniti, Yury Ivanenko

**Affiliations:** 1Laboratory of Neurobiology of Motor Control, Institute for Information Transmission Problems, Russian Academy of Sciences, 127951 Moscow, Russia; solopova@iitp.ru (I.A.S.); selionov@iitp.ru (V.A.S.); dolinsk-irina@yandex.ru (I.Y.D.); d.zhvansky@gmail.com (D.S.Z.); 2School of Biological and Medical Physics, Moscow Institute of Physics and Technology, 141701 Dolgoprudny, Russia; egor.blinov@phystech.edu; 3Laboratory of Neuromotor Physiology, IRCCS Santa Lucia Foundation, 00179 Rome, Italy; lacquaniti@med.uniroma2.it; 4Department of Systems Medicine and Center of Space Biomedicine, University of Rome Tor Vergata, 00133 Rome, Italy

**Keywords:** pattern generation circuitry, spinal cord neuromodulation, sensory stimulation, rhythmogenesis, H-reflex, human

## Abstract

**Simple Summary:**

Individual differences in the sensorimotor circuitry play an important role for understanding the nature of behavioral variability and developing personalized therapies. While the spinal network likely requires relatively rigid organization, it becomes increasingly evident that adaptability and inter-individual variability in the functioning of the neuronal circuitry is present not only in the brain but also in the spinal cord. In this study we investigated the relationship between the excitability of pattern generation circuitry and segmental reflexes in healthy humans. We found that the high individual responsiveness of pattern generation circuitries to tonic sensory input in both the upper and lower limbs was related to larger H-reflexes. The results provide further evidence for the importance of physiologically relevant assessments of spinal cord neuromodulation and the individual physiological state of reflex pathways.

**Abstract:**

The state and excitability of pattern generators are attracting the increasing interest of neurophysiologists and clinicians for understanding the mechanisms of the rhythmogenesis and neuromodulation of the human spinal cord. It has been previously shown that tonic sensory stimulation can elicit non-voluntary stepping-like movements in non-injured subjects when their limbs were placed in a gravity-neutral unloading apparatus. However, large individual differences in responsiveness to such stimuli were observed, so that the effects of sensory neuromodulation manifest only in some of the subjects. Given that spinal reflexes are an integral part of the neuronal circuitry, here we investigated the extent to which spinal pattern generation excitability in response to the vibrostimulation of muscle proprioceptors can be related to the H-reflex magnitude, in both the lower and upper limbs. For the H-reflex measurements, three conditions were used: stationary limbs, voluntary limb movement and passive limb movement. The results showed that the H-reflex was considerably higher in the group of participants who demonstrated non-voluntary rhythmic responses than it was in the participants who did not demonstrate them. Our findings are consistent with the idea that spinal reflex measurements play important roles in assessing the rhythmogenesis of the spinal cord.

## 1. Introduction

In the last decades, many researchers have put significant efforts into assessing the functional state of the spinal cord when performing locomotor movements [1,2,3,4,5]. It is generally accepted that neural control of stepping can be generated by spinal neuronal networks [6]. An important approach for promoting locomotor movements consists of the neuromodulation of the state of the central pattern generation (CPG) circuitry by spinal cord stimulation as well as by specific pharmacological neuromodulation [7,8,9]. For that reason, a large number of studies in recent years have been dedicated to developing various neuromodulation tools and testing them in both neurologically intact individuals and patients.

Normally, spinal CPGs are quiescent but their activity can be triggered by augmenting the input to the spinal cord. Experimentally, a manifestation of the rhythmic activity in humans can be facilitated under conditions in which the contribution of body weight and balance control is excluded (i.e., in the absence of external resistance). In particular, we have previously shown that tonic sensory stimulation (evoked by peripheral nerve electrical stimulation or by the vibrostimulation of muscle proprioceptors) can induce non-voluntary limb oscillations in non-injured subjects when their legs are placed in a gravity-neutral unloading apparatus (air-stepping) [5,10,11,12]. It is also worth mentioning that an inherent capacity for generating rhythmic movements can be observed both in the lower and upper limbs [12], likely because human locomotion involves rhythmic arm movements and shares many features with the forelimb and hindlimb coordination of quadrupeds [13].

However, while the presence of non-voluntary locomotor-like movements has been demonstrated hitherto in different laboratories and confirmed by using different stimulation techniques, large individual differences were observed in the responsiveness to such stimuli. The effects of sensorimotor neuromodulation manifest themselves only in some of the subjects. According to different studies, air-stepping can be evoked in ~10–50% of healthy subjects [5,10,11,14,15] and the degree of activation may depend on the state of the spinal cord. The excitability of the spinal circuitry may, in turn, depend on the balance of the supraspinal inhibitory and excitatory inputs that are directed to the spinal interneurons and the degree of inhibition of the afferent terminals on the motoneurons. It is worth noting that notable individual differences were also reported in the measurements of spinal segmental reflexes (e.g., the H-reflex) in neurologically intact individuals [16,17,18]. In fact, reflexes constitute an essential part of the locomotor program and may also be a sign of its functional state and impairment.

Individual differences in the sensorimotor circuitry play an important role for understanding the nature of behavioral variability [19,20] and they may also be essential for characterizing the state of the spinal circuitry and for providing markers of the individual response to different stimuli or treatments. For instance, there is a relationship between the facilitation of segmental reflexes (e.g., the H-reflex) and the ability to recover gait [21,22]. Given that the spinal cord is capable of sensory-induced functional plasticity, gait training has been shown to normalize H-reflex excitability in spinal rats [23] and in spinal cord injury patients [24,25,26]. As a step towards understanding the relationship between the responsiveness of the pattern generation circuitry and spinal segmental reflexes, we examined the manifestation of non-voluntary rhythmic movements that were evoked by muscle vibration in non-injured subjects, while the relative excitability of alpha motoneurons to excitatory inputs from Ia afferents was assessed using the electrical stimulation of the tibial and median nerves (soleus and flexor carpi radialis H-reflexes, respectively). We hypothesized that the excitability of the pattern generation circuitry and motoneurons to the sensory stimulation in neurologically intact individuals could be interrelated. To this end, we identified the participants who were and who were not responsive to tonic sensory stimulation and compared the H-reflex responses between these two groups of participants. Since the H-reflex is task-dependent and typically modulated during movements, in some experiments we applied stimuli at different phases of the cycle and compared the responses at similar amplitudes of angular motion and muscle activity. The results are discussed in the context of spinal cord reflex excitability and rhythmogenesis.

## 2. Materials and Methods

### 2.1. Participants

A total of thirty-three healthy volunteers (25 males, 8 females), aged between 23 and 70 years (mean 40 ± 3 years [mean ± SE]) participated in this study. Two experiments were performed. In the first experiment (Exp 1), we investigated non-voluntary upper limb rhythmic movements. The participants were 19 healthy volunteers (15 males and 4 females, aged 32 ± 3 years). In the second experiment (Exp 2), we investigated lower limb movements. The participants were 18 healthy volunteers (12 males and 6 females, aged 48 ± 4 years) and four of these subjects participated to Exp 1 as well. None of the subjects suffered from any known neurological or motor disorders. The experiments were performed according to the procedures of the Ethics Committee of the Institute for Information Transmission Problems (protocol n.14/11) and in conformity with the Declaration of Helsinki for experiments on humans. All of the subjects had given their informed consent. The total duration of each experiment was about 1.5 h.

### 2.2. Experimental Setup

The experimental setup for eliciting and recording the upper and lower limb non-voluntary cyclic movements was similar to that which has been described in our previous studies [5,12]. Even though we studied one-limb movements, the basic features of rhythmic movements are similar for one-leg and two-legged air-stepping [10]. To minimize the effects of gravity and external resistance, the subjects laid on their right side with a suspended arm (Exp 1, Figure 1A) or leg (Exp 2). When lying down, the subject’s relaxed, suspended limb assumed the equilibrium position with joint angles determined by the relative passive stiffness of the agonist and antagonists and other soft tissues around the joints. The trunk was fixed between two rests that were placed on the breast and back of the trunk in order to limit its rotation/tilt during limb movements, the other limbs were lying motionlessly and the head was put on a pillow. The subjects were instructed to relax and not intervene with movements that might be induced by stimulation.

### 2.3. Sensory Stimulation Technique

Continuous vibrostimulation (40–60 Hz sinusoid, ~1 mm amplitude) of m. triceps brachii (TB) in Exp 1 and m. quadriceps (Q) in Exp 2 was produced by the application of a small DC motor with an attached eccentric weight. We selected these muscles based on previous observations that air-stepping could be entrained by the vibrostimulation of the proprioceptors of these muscles [5,12]. The vibrator was placed in a cylindrical box (3 cm diameter, 7 cm length). Muscle vibration mainly stimulates Ia muscle spindle afferents, though other afferent signals may be elicited as well [27]. The vibrator was fastened with a rubber belt over the belly of the TB muscle (Exp 1) and over the Q muscle at the left knee tendon, about 5 cm over the patella (Exp 2). Prior to recording, one to two sensory stimulation probes were delivered so that the subjects could become familiar with the stimulus. For the test of the presence of non-voluntary limb oscillations, the duration of recording was 1 min (2–3 probes).

### 2.4. H-Reflex Measurements

The Hoffmann reflex measurements were performed in the flexor carpi radialis (FCR) muscle by stimulating the median nerve (Exp 1) and in the soleus muscle by stimulating the tibial nerve (Exp 2). The rationale for stimulating the median nerve innervating the hand muscles is that it might have access to the spinal pattern generation circuitry and play a part in the cervicothoracic cross-limb reflex modulation during stepping [13,18,28], while stimulation of the tibial nerve is commonly used for the soleus H-reflex assessment and its modulation during locomotor movements [26,29]. Hoffmann reflexes were elicited by delivering constant current square pulses (1 ms) to the median (Exp 1) or posterior tibial (Exp 2) nerve through bipolar surfaces electrodes (2 × 2 cm^2^ and an inter-electrode distance of 2 cm). For the FCR H-reflex, the bipolar stimulation electrodes were positioned over the median nerve (just below the ulnar fold) of the left (suspended) arm, firmly pressed with adhesive straps. For the soleus H-reflex, bipolar surfaces electrodes were placed in the popliteal fossa of the left (suspended) stationary leg, according to established protocols and methodologies [11]. 

Prior to the measurements, we determined the maximum M response amplitude (M_max_, using supramaximal stimulus intensity) and the intensity (3.5–10 mA) of ES for the H-reflex measurements was individually adjusted so as to obtain the M-wave of ~15% of M_max_. For the FCR H-reflex measurements (Exp 1), three conditions were used: stationary limbs, voluntary rhythmic movements and passive limb movements. For the soleus H-reflex measurements (Exp 2), the protocol was similar except that we measured the H-reflex only during stationary conditions. The stimulation procedure was the following: -First, in the stationary limb conditions (Exp 1 and 2), we applied ES of the median or tibial nerve every 2.5–5 s when the subject was relaxed and not stimulated (‘no vibration’) and during vibration. In the latter case, we applied ES after the onset of the vibration but in the absence of non-voluntary movements (normally, we could record 2–3 stimuli since the latency for evoking non-voluntary movements is about several seconds).-For voluntary movements (Exp 1), we asked the subject to perform rhythmic limb movements with a cycle duration of ~2 s (similar to that during non-voluntary rhythmic oscillations). The electrical stimulus was delivered to the median nerve every 2.5–5 s, which randomly occurred during different phases of the movements, so that we grouped and analyzed the H-reflexes during 4 phases of the movements (see below). The onset of the cycle was defined as the maximum shoulder extension.-For passive movements (Exp 1), the procedure was similar to that of ‘voluntary movements’ except that the experimenter imposed passive rhythmic (period ~2 s) movements of the suspended limb. The same experimenter performed passive movements and the frequency and angular amplitude of imposed movements in the upper limb joints were comparable for all subjects (see the Results).

About 2- to 3-min periods of rest were taken between these testing probes. For the FCR H-reflex measurements, the stationary conditions were repeated also after voluntary movements and after passive movements. 

### 2.5. Data Recording and Analysis

Recordings of the EMG activity were obtained by means of surface bipolar electrodes with the wireless Delsys Trigno EMG system (Delsys Inc., Boston, MA, USA), a bandwidth of 20–450 Hz and overall gain of 1000. EMG activity was recorded on the left side of the body from the following muscles: Exp 1—biceps brachii (BB), triceps brachii (TB), anterior (DA) and posterior (DP) deltoid muscles, flexor carpi radialis (FCR) and extensor carpi radialis (ECR); Exp 2—rectus femoris (RF), biceps femoris (BF, long head), soleus (SOL), lateral gastrocnemius (LG) and tibialis anterior (TA). Angular movements in the two joints of the arm (shoulder, elbow) and leg (hip, knee) were recorded by using potentiometers that were attached laterally to each joint [5,12]. The kinematic and EMG data were sampled at 1000 Hz.

The cycle duration and range of angular motion (ROM) were assessed by averaging the movement parameters across 10 cycles during sustained non-voluntary, voluntary and passive cyclic movements.

For the H-reflex analysis, the peak-to-peak amplitudes of the M-wave (over the 8–15 ms period after the stimulus in Exp 1 and 8–20 ms in Exp 2) and H-reflex (15–50 ms after the stimulus in Exp 1 and 25–60 ms in Exp 2) were automatically calculated online from each sweep. The M-waves and H-reflexes were normalized to M_max_ in order to reduce inter-subject variability. A custom-made constant-current stimulator provided the desired stimulation train and a computer program was used to trigger the electric stimuli. As mentioned above, the ES was individually adjusted in order to obtain the M-wave of ~15% of M_max_. As the muscle changes its length during air-stepping, the muscle fibers move relative to the recording EMG electrodes. Also, the stimulating electrode can move relative to the nerve. Therefore, the experimenter could slightly adjust (manually) the stimulus intensity for each cycle (since the stimulation time sequence was predefined in each trial) to achieve more similar M-waves across the stimulation times and conditions. In our analysis, we accepted only probes with the M-wave in the range 9–20% of M_max_ in Exp 1 and 10–21% of M_max_ in Exp 2 and the data with altered M-responses were discarded. After removing the sweeps with too large or small M-waves, the H-reflexes were averaged for each condition of each subject. About 5000 H-reflexes total were obtained from all of the subjects in Exp 1 and 400 H-reflexes for those in Exp 2. The background FCR and SOL activity was calculated over a 50 ms interval prior to the stimulus.

### 2.6. Statistics

The mean values and standard errors of the mean were used as descriptive statistics of the characteristics of the subjects and general movement parameters (ROM, cycle duration, mean amplitude of background EMG activity) that met the normal distribution criteria (Shapiro–Wilk’s W-test, *p* > 0.05). For reporting the percentage of the participants demonstrating non-voluntary upper and lower limb movements, the presence of oscillations was considered, if the amplitude of the cyclic movements in the shoulder or elbow joint exceeded 3°. For comparisons of the background EMG activity of the two participant groups, unpaired *t*-tests were used. The experimental data set of the H-reflex and M-responses did not meet the normal distribution criteria (Shapiro–Wilk’s W-test, *p* < 0.05), therefore non-parametric statistics were used. The descriptive statistics included medians, quartiles and range of values. For comparisons of two related groups, the Mann–Whitney test was used (effect size statistics: the relationship between f and the Mann–Whitney U (specifically U1) is as follows: f = U1/n1n2). The level of statistical significance was set at 0.05.

## 3. Results

### 3.1. Characteristics of Non-Voluntary Upper Limb Oscillations

The general characteristics and the occurrence of non-voluntary rhythmic arm oscillations (Exp 1) in 19 tested participants are reported in Figure 1B,C. All of the subjects were tested in the same unloading conditions in order to facilitate the manifestation of rhythmic patterns when the subject was laying on their side and proprioceptive stimulation was elicited by continuous TB muscle vibration. There were large inter-individual differences in the responsiveness to such sensory stimulation. In some subjects (e.g., subject 5, Figure 1B) we failed to elicit limb oscillations even if the vibrostimulation continued for more than 1 min. In other subjects, non-voluntary rhythmic movements could be elicited (e.g., subject 19, Figure 1B).

Figure 1C illustrates the amplitudes of the evoked shoulder and elbow oscillations for all 19 participants. In 9 subjects, vibration of the TB was effective in eliciting cycling arm movements. In 10 subjects, we failed to evoke non-voluntary limb oscillations. According to these responses, we divided the subjects into two groups: those who did not demonstrate and who demonstrated non-voluntary rhythmic arm oscillations (groups 1 and 2, respectively). Angular movements were observed in both shoulder and elbow joints, though the two joints were not involved to the same extent in all of the subjects: the ROM in the proximal (shoulder) joint was generally larger than that which was observed in the more distal (elbow) joint, so we ordered the participants according to the magnitude of the ROM in their shoulder joint (Figure 1C). Movements of the wrist joint were not recorded; however, they were normally absent (as assessed by visual observation by the experimenters). The latency of the elicited cyclic movements was several seconds (and varied across subjects), nevertheless, rhythmic arm movements over a period of ~2 s (1.8 ± 0.2 s [mean ± SE], Figure 1C) persisted as long as the stimulation continued, consistent with previous studies [12]. Non-voluntary arm oscillations could be accompanied by bursts of EMG activity (generally more prominent in the proximal muscles, DA and DP) with a predominantly reciprocal interaction between the flexors and extensors (Figure 1B, subject 19). Nevertheless, in most cases the induced EMG activity was very small or absent (median values well below 5 μV). However, it is worth noting that during arm swinging in normal upright walking the EMG activity of the upper limb muscles is also small [30]. Furthermore, air-stepping requires minimal muscle activity to produce cyclic movements due to the lack of external resistance when the limbs are placed in a gravity-neutral unloading apparatus [12,18] and since the movements are rather slow (~2 s cycle duration).

### 3.2. Assessments of the H-Reflex in the Upper Limb Muscle

In this section, we describe the characteristics of the H-reflex by comparing the responses between the two groups of participants. The subjects were tested in the same unloading conditions; namely, when the subject was laying on the side and their upper limb was suspended. For the H-reflex measurements, three conditions were used: stationary limbs, voluntary rhythmic movements and passive limb movements.

During the stationary conditions, we applied ES of the median nerve every 2.5–4 s when the subject was relaxed and not stimulated (the ‘no vibration’ condition) and during TB vibration. In the latter case, we applied ES after the onset of the vibration but in the absence of non-voluntary movements. If the experimenter visually detected the onset of movement, stimulation was stopped and the last stimulus during the movement was disregarded from the analysis. Typically, two to three stimuli were analyzed in group 2 prior to the appearance of arm movements (since the delay of non-voluntary limb oscillations was several seconds). Figure 2B illustrates the examples of the H-reflex in 2 subjects during the ‘no vibration’ and ‘TB vibration’ conditions. Note a significantly larger H-reflex in subject 15 than in subject 2 in the ‘no vibration’ condition.

Since the H-reflex is typically modulated during stepping, during the movement conditions we applied stimuli at different phases of the cycle and compared the responses at similar amplitudes of angular motion (see Methods). For voluntary movements, we asked the subject to perform rhythmic arm movements with a cycle duration of ~2 s (similar to that during the non-voluntary rhythmic oscillations, Figure 1C), while, during passive movements, the same experimenter performed similar upper limb oscillations (9 ± 2° for the elbow, 19 ± 3° for the shoulder). The electrical stimulus was delivered to the median nerve every 5 s, which randomly occurred during different phases of movements, so that we grouped and analyzed the H-reflexes during four phases of movements. The onset of the cycle was defined as the maximum shoulder extension (Figure 3A). Figure 3B,C illustrates the examples of the H-reflex measurements in four subjects during voluntary and passive movements, respectively. Note the larger H-responses in the subjects from group 2 (subj. 16 and 18) who demonstrated non-voluntary arm oscillations, than in the subjects from group 1 (subj. 2 and 4).

Figure 4 summarizes the overall results of the FCR H-reflex measurements in the two groups of subjects during stationary conditions, rhythmic voluntary movements and passive upper limb oscillations. While the M-responses were not significantly different between the two groups (*p* > 0.3, f = 0.56 in all of the conditions, since we adjusted the M-response to ~15% of M_max_), the H-reflex was on average significantly smaller for group 1 in the stationary conditions when the subject was relaxed (*p* < 0.02, f = 0.82, Mann–Whitney U test), though during TB vibration the difference was not significant (*p* = 0.23, f = 0.68, Mann–Whitney U test) (Figure 4A). During movement conditions, the extent of the modulation of the H-reflex throughout the movement cycle was much less (Figure 4B,C, in part because the EMG activity is very small during air-stepping) than during normal walking [29]. However, despite the lack of considerable modulation, the FCR H-reflex was significantly larger for group 2 participants during phases 1 and 2 of the voluntary movements (*p* < 0.01, f = 0.91 and *p* < 0.02, f = 0.85, for phases 1 and 2, respectively, Mann–Whitney U test, Figure 4B) and during all of the phases of passive cyclic movements (*p* < 0.009, f = 0.88 for phase 1, *p* < 0.009, f = 0.91 for phase 2, *p* < 0.01, f = 0.85 for phase 3 and *p* < 0.01, f = 0.85 for phase 4, Figure 4C, while the background EMG activity was similar, *p* > 0.4 for passive and voluntary movements, unpaired *t*-tests).

While the H-reflex was larger for group 2 subjects, we found only weak correlations between its value and the amplitude of the evoked non-voluntary (shoulder) oscillations (r = 0.48, 0.6 and 0.1 for the H-reflex values during ‘non vibration’, voluntary movements and passive movements, respectively), possibly because of a relatively small sample size of the participants in group 2 and/or due to individual differences in the facilitatory effects of the unloading conditions on the manifestation of rhythmic movements. Nevertheless, the results overall showed that the H-reflex was considerably higher in the group of participants who demonstrated non-voluntary rhythmic responses than it was in the participants who did not demonstrate them (Figure 2, Figure 3 and Figure 4).

### 3.3. Non-Voluntary Lower Limb Oscillations

In another experiment (Exp 2), we investigated the individual differences in the non-voluntary lower limb oscillations that are evoked by vibration of the Q muscle. An example of this evoked non-voluntary leg air-stepping movements is presented in Figure 5B (upper panel). Typically, the knee oscillations were larger than the hip oscillations (ankle movements were very small, though we did not record them) and they could be accompanied by EMG activity. In most subjects, TA, LG and RF showed very low activation levels (likely due to the lack of limb loading and absence of noticeable ankle joint movements), whereas rhythmic BF activity was often more prominent. As in the case of the upper limb stimulation, generally, cyclic movements started with a latency of several seconds and increased monotonically for 2–10 cycles (Figure 5B, upper panel) until they reached a relatively constant amplitude of angular oscillations (the coefficient of variation across 8 successive cycles was 8 ± 1% across all of the trials and subjects who demonstrated non-voluntary air-stepping movements) that persisted throughout the stimulus application. We averaged the amplitude and cycle duration of these oscillations for the last 8 cycles and have depicted them in Figure 5B (lower panels). The period of non-voluntary air-stepping movements (2.2 ± 0.2 s) was relatively long (compared with normal upright walking), likely due to the lack of limb loading and the effect of gravity on the pendulum-like behavior of the swinging limb when using a gravity-neutral unloading apparatus for air-stepping.

The vibration of Q was effective in evoking movements in 9 out of the 18 subjects. There were large inter-individual differences in the responsiveness to sensory stimulation and, as in the case of the upper limb movements, we divided the subjects into two groups (Figure 5B): those who did not demonstrate and those who demonstrated non-voluntary rhythmic leg oscillations (groups 3 and 4, respectively). Most of the participants in Exp 2 were different than those who participated in Exp 1. For the 4 subjects who participated in both experiments (though on different days), their results were relatively consistent: one subject demonstrated non-voluntary movements in their both upper and lower limbs, two subjects did not show non-voluntary movements and only one subject showed rhythmic responses in their arms but not in their legs.

### 3.4. Soleus H-Reflex

Following the probes with Q vibration, we performed the measurements of the soleus H-reflex in all of the participants and compared these measurements between the two groups. For the lower limb protocol, we measure the H-reflex in the stationary conditions, similar to those that were used for the upper limb experiments: namely, at rest (no vibration) and during Q vibration (but prior to the onset of non-voluntary air-stepping). The M-response amplitudes were similar in the two groups (*p* > 0.1, f = 0.53, Mann–Whitney U test) and the background EMG activity was also similar (*p* > 0.3, unpaired *t*-test) (it was minute, if any, in the stationary legs).

Figure 5C illustrates the examples of the H-reflex measurements in three subjects of group 3 and group 4. Variability in the H-reflexes was notable for each group and the correlations between the H-reflex values in the stationary conditions and the amplitude of the non-voluntary leg movement (knee joint ROM) in group 4 participants were relatively small (r = 0.27 and r = 0.39 for the H-reflex values during the ‘no vibration’ and ‘vibration’ conditions), as in Exp 1. Nevertheless, despite some inter-individual variability and ‘overlap’ between the individual values of the two groups, the H-reflex was significantly larger for group 4 participants during both the ‘no vibration’ condition (*p* = 0.01, f = 0.85, Mann–Whitney U test) and during ‘Q vibration’ (*p* = 0.005, f = 0.92) (Figure 5D).

### 3.5. Effect of Age

There could be various factors determining the individual sensorimotor responses (both the H-reflex and manifestation of non-voluntary movements) such as, for instance, the participants’ age (range 23–70 years). However, we did not find significant age differences between the two related groups of subjects (those who showed and those who did not show the non-voluntary rhythmic movements; *p* = 0.1 in Exp 1 and *p* = 0.6 in Exp 2, unpaired *t*-tests). There were also low correlations between the H-reflex values and the age of the participants (e.g., r = −0.23 and r = −0.1 for stationary conditions in Exp 1 and Exp 2, respectively). Possibly, a larger sample of subjects is needed to investigate the effect of age and/or the other factors that can contribute to the observed individual differences in the responsiveness of the pattern generation circuitry to sensory stimulation.

## 4. Discussion

In order to investigate the relationship between the excitability of the pattern generation circuitry and segmental reflexes in healthy humans, we applied sensory tonic stimuli that have previously been shown to be effective in eliciting automatic locomotor-like movements during unloading conditions and we also recorded the H-reflex at rest and during rhythmic movements. Given that human locomotion involves rhythmic arm movements and that an inherent capacity for generating rhythmic movements can be observed both in the lower and upper limbs, we studied these relationships for both arm and leg oscillations. The presence of rhythmic stepping responses and their relation to the H-reflex was assessed in 33 neurologically intact individuals. The common feature of these findings is that the high individual responsiveness of pattern generation circuitries to tonic sensory input was frequently associated with the larger H-reflexes in the upper and lower limb muscles (Figure 2, Figure 3, Figure 4 and Figure 5).

### 4.1. Methodological Considerations

There are some limitations for interpreting the results of the H-reflex measurements and several factors may shape individual responses, such as individual differences in the reflex recruitment curve or our inability to distinguish the relative contribution of each spinal inhibitory mechanism to motoneuronal excitability during a specific motor task [31]. Nevertheless, the H-reflex remains one of the major available probes for investigating the excitability and adaptation of interneuronal networks in health and pathology, even though it characterizes only a part of the neuronal pathways [17,22,31,32].

In order to match the conditions and decrease the variability in the H-reflex amplitude, the M-wave amplitudes were maintained at ~15%M_max_ and were not significantly different between the groups of subjects and conditions (Figure 4A and Figure 5D, right panels), confirming the stimulus’ constancy. The reflex magnitude can change substantially during the contraction or stretching of agonist and antagonist muscles. However, typically low level (if any) background activity was observed in the FCR and SOL muscles during the experiments and this was not significantly different across the related groups of subjects (as assessed in the 50-ms interval prior to the stimulus). For the voluntary and passive movement conditions (Figure 3 and Figure 4B), we grouped the H-responses according to the phase of movement a posteriori, based on the criteria of the stimulus falling into the appropriate interval after the onset of the movement cycle (Figure 3A). Nevertheless, it is unlikely that the more precise timing of the stimulation in each of those four phases would change the general conclusion (the significant difference between the two related groups of subjects) since there was little modulation of the FCR H-reflex across the four phases of voluntary and passive movement (Figure 4B, left and right panels, respectively). Overall, the measurements showed significantly higher H-reflex amplitudes in the subjects who demonstrated non-voluntary rhythmic limb oscillations in response to vibrostimulation of the muscle proprioceptors. Furthermore, similar results were obtained for the different conditions (‘no vibr’, ‘vibr’, voluntary and passive movements) and for the two separate experiments (Exp 1 and 2, in which mostly different subjects participated), suggesting that the responsiveness of pattern generation circuitries to tonic sensory input is commonly related to the larger H-reflex amplitude in the upper and lower limb muscles.

### 4.2. Spinal Cord Plasticity

While the spinal circuitries likely require a more rigid organization compared to the highly flexible cortical circuits due to their proximity to the outer world [33], nevertheless it becomes increasingly evident that the spinal cord is adaptable and is not a simple relay structure in mediating the interplay between the spinal and supraspinal networks. It holds many features of the supraspinal circuitries, such as activity-dependent homeostasis, the turnover of synapses (the half-life of synaptic proteins can be on the order of hours to weeks [34]), structural functional units [35], etc. As a first step in the chain of sensory processing, the spinal cord is already capable of performing transformations that map information from sensory coordinates to motor coordinates and encode a global representation of limb mechanics and thus it contains a reduced form of the body schema [36,37,38,39]. Plastic changes in the motoneurons, interneurons and sensory afferent terminals have been documented in a large number of studies [33]. Inhibitory synapses prevail in the spinal cord [40] and serve to maintain the network’s stability, appropriate task-dependent tuning of reflexes and activation of neurons of a particular spinal functional unit. It is worth noting that the same motoneuron and interneuron can participate in a vast repertoire of possible movements. These considerations raise important questions about appropriate selection and tuning of spinal functional units and sensory feedback [35]. Furthermore, recent intriguing findings suggest an extensive amount of redundancy among spinal locomotor circuits that enables the selection of many combinations of neurons (synapses) when generating each locomotor cycle [41,42]. The full consideration of the spinal cord’s plasticity and functioning is likely out of the scope of this study; however, it is important to stress that inter-individual variability and adaptability in the neuronal circuitry is present not only in the brain but also in the spinal cord.

### 4.3. Inter-Individual Variability in the H-Reflex and Neuromodulation of the Spinal Cord

One important property of the spinal circuitry is its capacity to generate and maintain rhythmic movements; locomotor activity requires an appropriate tonic excitatory state of the spinal pattern generation circuitry [3]. The tonic excitatory input is an integral part of many CPG models that have been developed to explain the functioning of spinal pathways for reciprocal motoneuron activation. Experimentally, automatic stepping movements in humans can be evoked when spinal locomotor circuits are directly activated by relatively nonspecific stimuli such as continuous muscle vibration [10], tonic electrical stimulation of the peripheral nerves [5], direct electromagnetic stimulation of the spinal cord [14] or epidural or transcutaneous electrical stimulation [24,43,44]. All of these techniques that facilitate the triggering of locomotor-like movement are based on the stimulation of a part (depending on the technique) of the large-diameter afferent fibers entering the spinal cord through the dorsal roots, which spread their activation and increase the excitability of several segments of the spinal cord. Afferent stimulation was also probably the first method for producing rhythmic movements in animal preparations [45].

The potential reasons behind the inter-subject variabilities in activating the pattern generation circuitry by tonic sensory stimulation could be related to differences in the physiological state of the CPG circuitry and the extent of descending inhibition. The fact that only some of the subjects demonstrated relatively robust locomotor-like activity in response to tonic sensory stimulation is likely related to the specific individual excitatory state of the spinal cord. For instance, in most of the subjects who participated in both of the experiments, the results (i.e., the presence/absence of non-voluntary movements) were consistent for the upper and lower limbs. Also, it has been previously reported that subject selectivity in locomotor responses could not be attributed only to experimental variability, because every subject who demonstrated a response to stimulation during the first testing session demonstrated a similar response during every subsequent testing session, suggesting different levels of supraspinally mediated inhibition of the spinal circuitry from subject to subject [14]. An interesting feature of the above-mentioned findings is that continuous sensory stimulation does not produce tonic vibratory reflexes or static changes in the configuration of the suspended limbs. Instead, it produces air-stepping, consistent with a general notion that non-specific activation of the spinal cord may elicit rhythmic limb oscillations [3]. However, even for the tonic vibratory reflex, the effects of muscle vibrostimulation manifest themselves only in some of the subjects [46]. The implications of the selective nature of the generation of stepping-like activity are interesting and could reflect a general property of tonogenic structures.

In this study, we investigated inter-individual variability in the responsiveness of pattern generation circuitry to sensory stimulation in healthy subjects in the context of looking for potential individual ‘signatures’ of its state. The results were quite clear: despite some ‘overlap’ in the H-reflex measurements of the related groups, the H-reflex was significantly larger in the participants who showed non-voluntary limb oscillations in response to tonic sensory stimulation. It is worth noting that this result has been obtained for both the upper and lower limbs (Figure 4 and Figure 5) and for different (stationary and movement) conditions of the H-reflex’s measurement. In the case of the dynamic conditions, the lack of substantial phase modulation of the FCR H-reflex during the passive and voluntary movements (Figure 4B,C) could be explained by the relatively low muscle activity and its modulation, by the reduced contribution from central motor commands during the passive movements, or by a relatively low resolution (only 4 phases of stimulation) [28]. In conjunction with previous studies in which it was found that FCR H-reflex amplitude during walking was suppressed by 20% compared to that which was measured during static conditions [28], we also obtained somewhat suppressed H-reflex (cf. H-reflex values during the ‘no vibr’ and voluntary/passive movements, Figure 4). Nevertheless, when compared in the same static or dynamic conditions, the larger H-reflex amplitudes were consistently observed in the groups of subjects with the high responsiveness of their pattern generation circuitry to tonic sensory input (Figure 4 and Figure 5).

The relationship between the excitability of pattern generation circuitry and segmental reflexes is likely to reveal some characteristic properties of the functioning of the spinal cord circuitry. Sensory stimulation facilitates the expression of locomotion, permits its adaptation to the environment, promotes transitions between different phases of the locomotor cycle and helps to induce use-dependent plasticity, which is a hallmark of physical therapy treatment [47]. The identification of the pathways that are capable of influencing the excitability of rhythm-generating modules allows physiologically relevant assessments and can reveal potential therapeutic targets [7,48]. Although the active engagement of reflex operant conditioning has been used to rehabilitate locomotor function [22], the extent to which spinal reflexes are related to the rhythmogenesis of the spinal cord has not previously been exploited. The current data do not allow the differentiation of the contribution of the descending and specific spinal inhibitory pathways on the state of the pattern generation circuitry; however, they strongly support the relationship between the neuromodulation of the spinal cord and the individual physiological states of the reflex pathways.

## 5. Conclusions

Individual differences in the sensorimotor circuitry play an important role for understanding the nature of behavioral variability [19,20] and developing personalized therapies [26,49,50]. As it is commonly studied for cortical and subcortical network plasticity and specific individual alterations, it is also reasonable to carefully consider and investigate the individual differences in the state of the spinal cord. The rationale is also the following. If the state of the spinal circuitry is tuned differently or impaired, it should be controlled differently by descending motor pathways, which, in turn, would change the involvement of the supraspinal structures in order to control the spinal neurons. These reciprocal spinal–supraspinal compensatory mechanisms may even create a risk of irreversible changes in the state of the locomotor circuitry in some pathologies, for instance during early development [51,52,53]. These considerations might be important both for characterizing individual motor skills and behavioral variability and for developing central pattern generator-modulating therapies in the case of spinal cord injuries. The results provide further evidence that spinal reflex measurements play important roles in assessing spinal cord rhythmogenesis and stimulate further research on looking for the individual ‘signatures’ of the spinal cord’s functioning.

## Figures and Tables

**Figure 1 biology-11-00707-f001:**
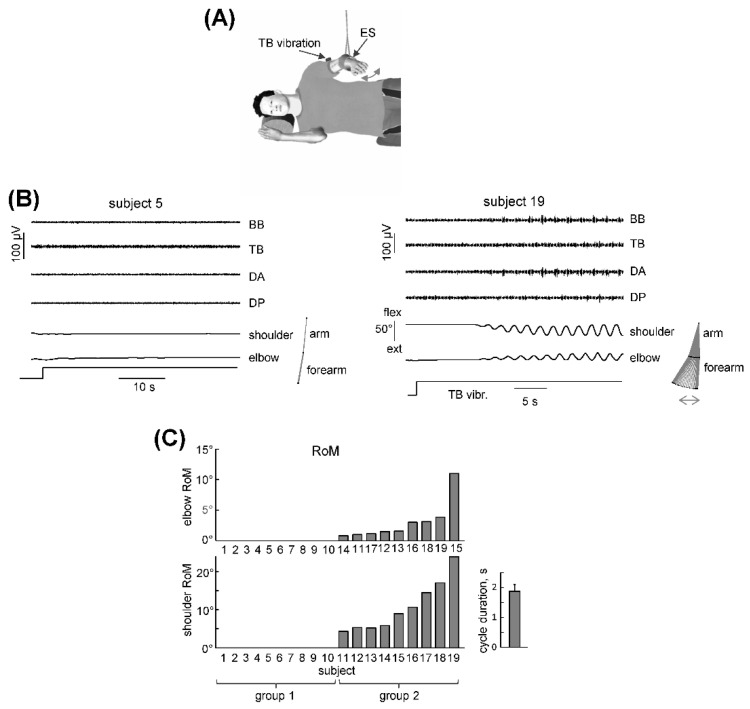
General characteristics of non-voluntary rhythmic upper limb movements. (**A**)—experimental setup for investigating arm rhythmic movements and their relationship with the relative excitability of alpha motoneurons to excitatory inputs from Ia afferents (H-reflex). Muscle vibration was used to evoke non-voluntary limb oscillations while electrical stimulation (ES) of the median nerve was used to evoke the H-reflex. The legs and the right arm were stationary. (**B**)—examples of changes in limb kinematics and EMG activity in response to continuous TB (triceps brachii) vibration in two subjects. BB—biceps brachii, TB—triceps brachii, DA—deltoideus anterior portion, DP—deltoideus posterior portion. Right panels show stick diagrams of upper limb oscillations. Note the absence of limb movements in subject 5 and their appearance in subject 19. (**C**)—range of angular motion (ROM) of evoked elbow and shoulder joint oscillations in all individual subjects (ordered by the magnitude of the ROM in the shoulder joint). The subjects were divided into two groups: with and without rhythmic arm movements (groups 2 and 1, respectively). Cycle duration (mean + SE) of evoked movements is also shown for group 2 (right panel).

**Figure 2 biology-11-00707-f002:**
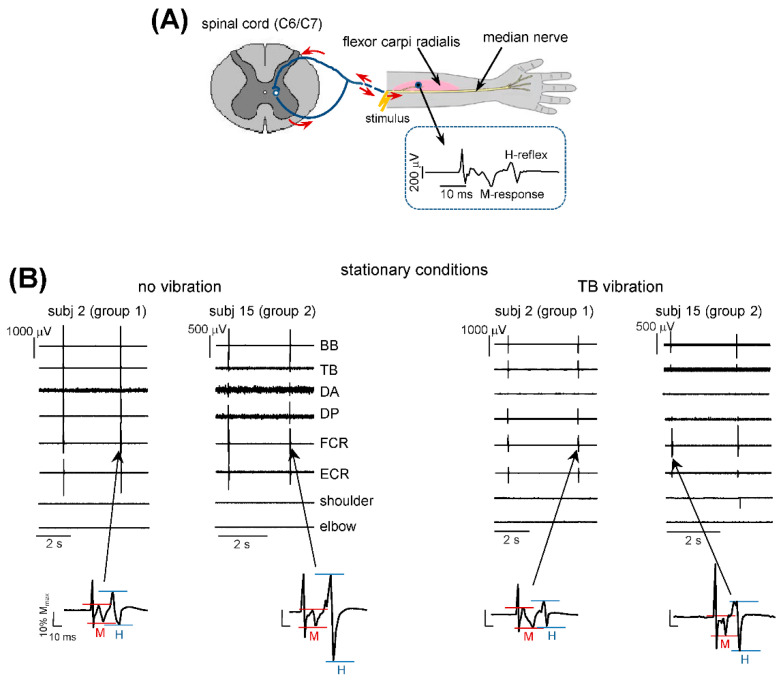
H-reflex measurements. (**A**)—H-reflex was measured in the flexor carpi radialis muscle during electrical stimulation of the median nerve. (**B**)—examples of waveforms of the H-reflex in two subjects (subj. 2 and 15 belonging to groups 1 and 2, respectively, see Figure 1C) during stationary conditions: left panels—no vibration, right panels—TB (triceps brachii) vibration. In the latter case, 2 stimuli were delivered after the onset of vibration but prior to the appearance of non-voluntary arm movements (note the absence of noticeable EMG activity in all muscles). BB—biceps brachii, TB—triceps brachii, DA—deltoideus anterior portion, DP—deltoideus posterior portion, FCR—flexor carpi radialis, ECR—extensor carpi radialis. Lower panels show corresponding waveforms of the M and H responses. The magnitude of M and H responses was marked by the red and blue horizontal lines, respectively. Note a significantly larger H-reflex in subj. 15 than in subj. 2 during the ‘no vibration’ condition.

**Figure 3 biology-11-00707-f003:**
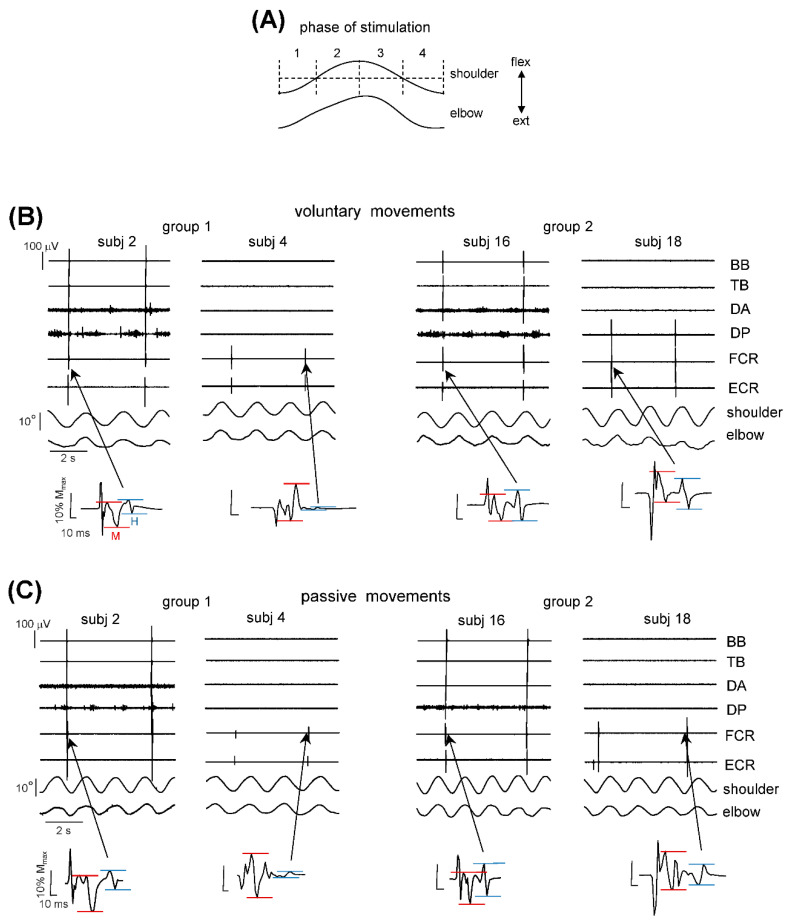
H-reflex measurements during rhythmic movements. (**A**)—phases (1–4) of stimulation (the onset of the cycle was defined as the maximum shoulder extension). (**B**)—examples of the H-reflex measurements during voluntary rhythmic movements in the subjects from group 1 (subj. 2 and 4) and group 2 (subj. 16 and 18). Lower panels show corresponding waveforms of the M- and H-responses (same format as in Figure 2B). (**C**)—examples of the H-reflex measurements in the same subjects during passive rhythmic movements. The cycle duration of voluntary and imposed passive rhythmic movements (~2 s) was maintained similar to that during involuntary rhythmic oscillations (Figure 1C). The stimulus was delivered every 5 s (which randomly occurred during different phases of movements). Note typically larger H-responses in the subjects from group 2 (subj. 16 and 18) who demonstrated non-voluntary arm oscillations. BB—biceps brachii, TB—triceps brachii, DA—deltoideus anterior portion, DP—deltoideus posterior portion, FCR—flexor carpi radialis, ECR—extensor carpi radialis.

**Figure 4 biology-11-00707-f004:**
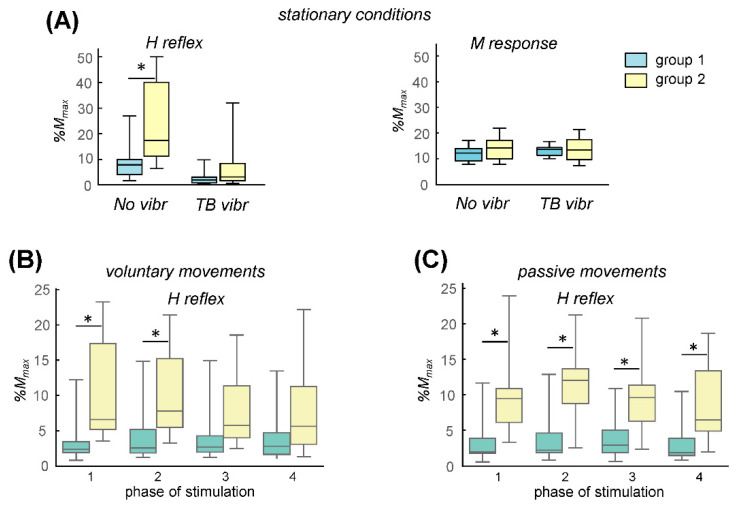
H-reflex amplitudes (median and quartiles) for the two groups of subjects measured during stationary conditions (**A**)—rhythmic voluntary arm movements, (**B**)—passive upper limb oscillations, (**C**)—in the former case (panel A), two stationary conditions were analyzed: no vibration and TB (triceps brachii) vibration. Corresponding M-responses are also plotted in the right panel. In the latter cases (panels B and C), H-reflex measurements were compared during 4 phases of movements (Figure 3A). Asterisks denote significant differences between the two groups.

**Figure 5 biology-11-00707-f005:**
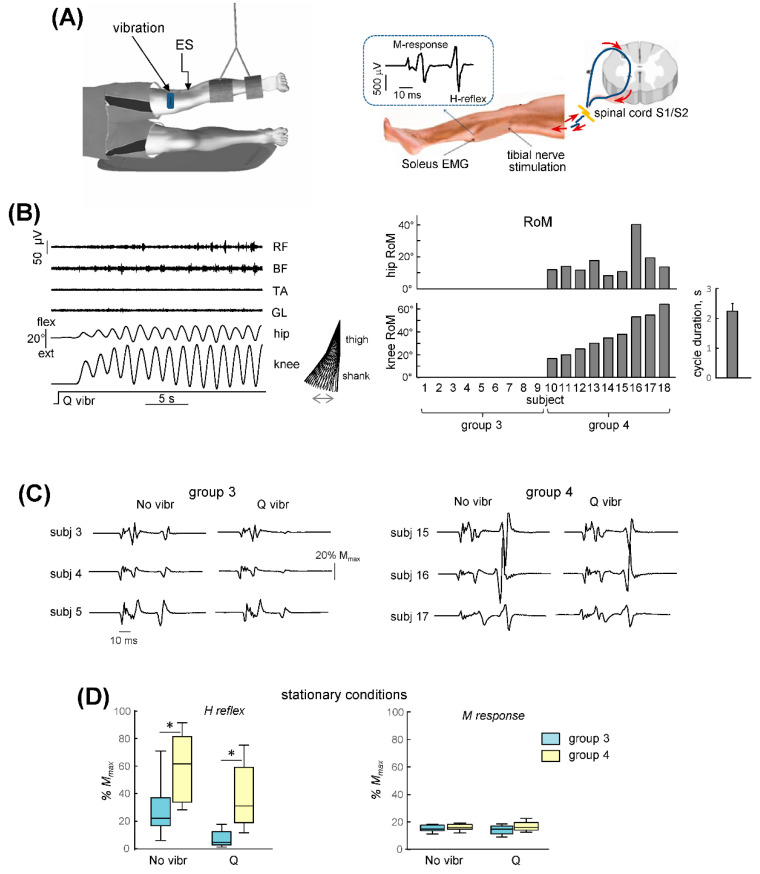
General characteristics of non-voluntary rhythmic leg air-stepping movements and H-reflex evaluation. (**A**)—experimental setup and soleus H-reflex measurements. Quadriceps (Q) muscle vibration was used to evoke non-voluntary limb oscillations, while ES of the tibial nerve was used to evoke the H-reflex. The arms and the right leg were stationary. (**B**)—left panel illustrates an example of evoked movements and EMG activity in the subject who demonstrated prominent non-voluntary limb oscillations and right panels show the ROM of hip and knee oscillations in all individual subjects (ordered by the magnitude of the ROM in the knee joint). The subjects were divided into two groups: with and without rhythmic leg movements (groups 4 and 3, respectively). Cycle duration (mean + SE) of evoked oscillations is also shown for group 4. RF—rectus femoris, BF—biceps femoris, TA—tibialis anterior, LG—lateral gastrocnemius. (**C**)—examples of the H-reflexes in subjects from group 3 (subj. 3, 4, 5) and group 4 (subj. 15, 16, 17) during stationary conditions: no vibr. and Q vibration. In the latter case, 2 stimuli were delivered after the onset of vibration but prior to the appearance of non-voluntary leg movements (as in the case of upper limb experiments, Figure 2B). (**D**)—H-reflex amplitudes (median and quartiles) and M-responses for the two groups of subjects. Asterisks denote significant differences between the two groups.

## Data Availability

Data are available upon reasonable request.

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
