# Peer review of "Higher Responsiveness of Pattern Generation Circuitry to Sensory Stimulation in Healthy Humans Is Associated with a Larger Hoffmann Reflex"

_biology, 2022, doi:10.3390/biology11050707_

Round 1

Reviewer 1 Report

“Higher responsiveness of pattern generation circuitry to sensory stimulation in healthy humans is associated with the larger H-reflex”

Overall strengths of the article:

This is a very interesting manuscript showing the relationship between the excitability of pattern generation circuitry and segmental reflexes in healthy humans. The spinal reflex measurements play important roles in assessing spinal cord rhythmogenesis and functioning. As the spinal reflexes are an integral part of the neuronal circuitry, here authors have investigated the extent to which spinal pattern generation excitability in response to vibrostimulation of muscle proprioceptors can be related to the H-reflex magnitude, in both limbs. They observed that the high individual responsiveness of pattern generation circuitries to tonic sensory input in both upper and lower limbs was related to the larger H-reflexes. These results provide further evidence for the importance of physiologically relevant assessments of spinal cord neuromodulation and the individual physiological state of reflex pathways. As a step toward understanding the relationship between the responsiveness of pattern generation circuitry and spinal segmental reflexes. These considerations might be important both for characterizing individual motor skills and behavioral variability and for developing central pattern generator-modulating therapies in case of spinal cord injuries. Overall this manuscript is well written and easy to follow. However, this paper suffers from some major limitations that should be addressed before publication. Details are in the specific comments section.

Specific comments on weaknesses:

  1. The descending inhibition plays an important role in reshaping the spinal segmental reflexes, I think they should include that in the discussion.
  2. The study aims and the directly corresponding study hypotheses must be clearly stated in the Introduction section.
  3. 1 A, B, & C should be either left to right or top to bottom, it looks like subject 19 is part of fig 1C. Abbreviations (e.g. BB, TB, …) used should be explained in the figure legend. Similarly in the other figs.
  4. Line-238; “There were large inter-individual differences in the responsiveness to such sensory stimulation. In some subjects (e.g., subject 5, Fig. 1B), we failed to elicit limb oscillations even if vibrostimulation continued for more than 1 min. In other subjects, non-voluntary rhythmic movements could be elicited (e.g., subject 19, Fig. 1B).” also Line-244; “In 10 subjects, we failed to evoke non-voluntary limb oscillations.” The potential reasons behind these inter-subject variabilities must be discussed.
  5. References need to be formatted carefully.

Author Response

We thank the reviewer for his/her evaluation of our study and comments that served to improve the manuscript, we tried to incorporate them all. Changes in the manuscript are marked in red.

Specific comments on weaknesses:

  1. The descending inhibition plays an important role in reshaping the spinal segmental reflexes, I think they should include that in the discussion.

We thank the reviewer for this suggestion. Indeed, as we mentioned in the Introduction (ln 67), “The excitability of the spinal circuitry may, in turn, depend on the balance of supraspinal inhibitory and excitatory inputs directed to spinal interneurons and the degree of inhibition of afferent terminals on motoneurons.” As suggested, we now included these considerations in the Discussion (p13):

“The potential reasons behind inter-subject variabilities in activating pattern generation circuitry by tonic sensory stimulation could be related to differences in the physiological state of the CPG circuitry and the extent of descending inhibition. The fact that only a part of subjects demonstrates relatively robust locomotor-like activity in response to tonic sensory stimulation is likely related to specific individual excitatory state of the spinal cord. For instance, in most subjects who participates in both ex-periments the results (i.e., the presence/absence of non-voluntary movements) were consistent for upper and lower limbs. Also, it has been previously reported that subject selectivity in locomotor responses could not be attributed only to experimental variability, because every subject that demonstrated a response to stimulation during the first testing session demonstrated a similar response during every subsequent testing session, suggesting different levels of supraspinally mediated inhibition of the spinal circuitry from subject to subject [14]. An interesting feature of the above-mentioned findings is that continuous sensory stimulation does not produce tonic vibratory re-flexes or static changes in the configuration of the suspended limbs. Instead, it produces air-stepping, consistent with a general notion that non-specific activation of the spinal cord may elicit rhythmic limb oscillations [3]. However, even for the tonic vibratory reflex, effects of muscle vibrostimulation manifest themselves only in part of the subjects [46]. The implications of the selective nature of the generation of stepping-like activity are interesting and could reflect a general property of tonogenic structures.”

  1. The study aims and the directly corresponding study hypotheses must be clearly stated in the Introduction section.

We added (ln 86):

We hypothesized that the excitability of the pattern generation circuitry and motoneurons to sensory stimulation in neurologically intact individuals could be interrelated. To this end, we identified the participants who were and who were not responsive to tonic sensory stimulation and compared the H-reflex responses between these two groups of participants.

  1. 1 A, B, & C should be either left to right or top to bottom, it looks like subject 19 is part of fig 1C. Abbreviations (e.g. BB, TB, …) used should be explained in the figure legend. Similarly in the other figs.

We adjusted Fig 1C, and similarly Fig.5A,B, and we also explained abbreviations of the muscles in all figure legends, as suggested.

  1. Line-238; “There were large inter-individual differences in the responsiveness to such sensory stimulation. In some subjects (e.g., subject 5, Fig. 1B), we failed to elicit limb oscillations even if vibrostimulation continued for more than 1 min. In other subjects, non-voluntary rhythmic movements could be elicited (e.g., subject 19, Fig. 1B).” also Line-244; “In 10 subjects, we failed to evoke non-voluntary limb oscillations.” The potential reasons behind these inter-subject variabilities must be discussed.

As suggested, we discussed the potential reasons behind inter-subject variability in activating pattern generation circuitry by tonic sensory stimulation (ln 529-548, see also our response to point 1).

  1. References need to be formatted carefully.

We used Zotero to automatically format the references according to the MDPI style. We now checked again the instructions for authors and the reference formatting guide (see https://www.mdpi.com/authors/references) and it looks like we correctly cited the articles. Or perhaps we missed some information or the reviewer noticed some specific errors? We would be happy to fix them. Likely, the publisher is supposed to finally adjust the references by adding the hyperlinks to references.

Reviewer 2 Report

The authors presented the spinal pattern generation excitability in response to vibrostimulation of muscle proprioceptors, and  to which extent can be related to the H-reflex magnitude, in both lower and upper limbs. 

Only a  minor comment: Please insert a full word in a title instead a "H"  -reflex.

Author Response

We thank the reviewer for his/her evaluation of our study.

Only a minor comment: Please insert a full word in a title instead a "H"  -reflex.

done

Reviewer 3 Report

This manuscript is well written. I think it's suitable for publication.

Author Response

We thank the reviewer for his/her evaluation of our study and manuscript.

Reviewer 4 Report

In the first sentence, the last decades should be clearly stated.

Second paragraph, a comma should follow "Normally,"

the use of i.e., should be in parentheses (i.e., ...).

Line 67 may in turn should appear as ", may in turn,"

The method section subheading "Subjects" should be changed to "Participants"

The statistics should report the mean and standard error of the mean rather than deviation.

The Mann Whitney U statistics are not reported correctly in the  results section. This is true for all statistics as using the p-value alone does not clarify which statistic is being described for each experiment throughout the manuscript.

Otherwise, the rest of the manuscript is clear, informative, and well written.

Author Response

In the first sentence, the last decades should be clearly stated.

We replaced “It the last decades” with “In the last decades”

Second paragraph, a comma should follow "Normally,"

done

the use of i.e., should be in parentheses (i.e., ...).

done

Line 67 may in turn should appear as ", may in turn,"

done

The method section subheading "Subjects" should be changed to "Participants"

done

The statistics should report the mean and standard error of the mean rather than deviation.

As suggested, we replaced SD with SE throughout the manuscript (including Fig. 1C and Fig.5B).

The Mann Whitney U statistics are not reported correctly in the results section. This is true for all statistics as using the p-value alone does not clarify which statistic is being described for each experiment throughout the manuscript.

In the original submitted manuscript we indicated in the “Statistics” session whish statistics has been used for specific parameters. However, we agree that in the Results there could be a lack of information at places, therefore we now indicated which statistics is being described for each experiment throughout the Results, as suggested by the reviewer.

Otherwise, the rest of the manuscript is clear, informative, and well written.

We thank the reviewer for his/her evaluation of our study and comments that served to improve the manuscupt.

Round 2

Reviewer 1 Report

This study addresses an important question towards understanding the relationship between the responsiveness of pattern generation circuitry and spinal segmental reflexes. This is important both for characterizing individual motor skills and behavioral variability following SCI and for developing central pattern generator-modulating therapies, a long-standing problem for which no satisfactory solution has been found yet. In the revised manuscript authors have successfully addressed all the comments raised by the reviewer and incorporated all the suggestions to improve the quality of the paper. I think this manuscript has been sufficiently improved from the previous version.